# Growth and Productivity of Sweet Cherry Varieties on Hungarian Clonal *Prunus mahaleb* (L.) Rootstocks

Károly Hrotkó [1,*], Krisztina Németh-Csigai [2], Lajos Magyar [1] and Gitta Ficzek [3]

1    Department of Floriculture and Dendrology, Hungarian University of Agriculture and Life Sciences, Villányi Str. 35-43, 1118 Budapest, Hungary
2    County Government Office, Tusnádi Str. 4-6, 9028 Győr, Hungary
3    Department of Fruit Growing, Hungarian University of Agriculture and Life Sciences, Villányi Str. 35-43, 1118 Budapest, Hungary
*    Correspondence: hrotko.karoly@uni-mate.hu

**Abstract:** Due to climate changes, drought- and lime-tolerant *Prunus mahaleb* rootstock may gain importance. Among the Mahaleb rootstocks and hybrids, there are standard and moderate-vigorous types, but their intensive testing in orchards is still needed. Our paper reports on testing rootstocks SL 64, Bogdány, Magyar, SM 11/4 clonal Mahalebs, and the hybrid MaxMa 14. 'Carmen', 'Vera', 'Paulus', and 'Rita' sweet cherry trees were trained on the above rootstocks to the principles of Hungarian Cherry Spindle at a spacing of 1.6 × 5 m. Rootstocks SL 64, Bogdány, and SM 11/4 proved to be vigorous, while on rootstocks Magyar and MaxMa 14, the trees were moderately vigorous, about 80%. 'Carmen', 'Vera', and 'Rita' on Magyar and MaxMa 14 produced high cumulative yields without significant differences, while 'Paulus' trees were most productive on Bogdány rootstock. 'Carmen' on Bogdány rootstock, 'Vera' on Magyar and Maxma 14 rootstock, and 'Rita' on MaxMa 14 were more precocious than on SL 64. Contrary to SL 64 and MaxMa 14, both Magyar and Bogdány rootstocks resulted in abundant flat branching and good fruit size. Our conclusion is that trees on Magyar and Bogdány rootstocks fit well to the Hungarian Cherry Spindle orchard system with 1250 tree/ha orchard density.

**Keywords:** branching habit; cumulative yield; fruit quality; high density orchard; precocity



## 1. Introduction

*Prunus mahaleb* is a major cherry rootstock in central and southern European countries, in Asia Minor, Central Asia, and northwest China [1–6]. The adaptability of the Mahaleb cherry to the continental climate, tolerance to drought, hot summers, poor soil, and lime will increase its importance to cherry growing in the future [5,7–10]. These rootstock traits will gain importance due to accelerated climate changes and replant conditions in western European countries, too [11]. Despite all of the advantages of *P. mahaleb*, very little breeding work has been conducted to improve its rootstock traits, probably because of its growth vigor. The Mahaleb rootstock selection of the last century mainly consists of seed tree genotypes and a few clonal rootstocks [5]. The recently released vegetative propagated *P. mahaleb* clones show some extended vigor range, less vigor with considerable precocity, and are uniform both in the nursery and in the orchard [12–15], but their adaptability in high-density orchard conditions is still in doubt or not accepted.

There is no doubt that in modern orchard systems, so-called "pedestrian orchards" of sweet cherry, growth-controlling and precocious rootstocks provide large advantages. Growth-reducing rootstocks allow plantation density of up to 1000–5000 trees/ha [16–18]. However, the modern dwarfing rootstocks widely planted in intensive orchard systems require optimal site conditions. In several site conditions in southern and southeastern European countries or Central Asia and China (high pH, calcareous soil, drought, hot summers), medium-vigorous or vigorous cherry rootstocks proved to be more applicable

than dwarfing ones [3,4,6,7,18–21], where the growth control could be supported by the application of summer pruning, pruning of the roots, and water restriction [4,5,7,21]. Even under preferable conditions due to climate changes and the more frequent replant situation, growers may need more robust rootstocks [11]. In addition to rootstocks, growth control in spindle trees or open center canopy (Spanish bush) on vigorous rootstocks could be managed by the application of frequent summer pruning, root-pruning, and water restriction [1,4,21].

Adaptability to different soil conditions is an important rootstock trait. *Prunus mahalebs* and its derivatives are best suited to light sandy or gravely soil types with free drainage, and they tolerate high lime content and pH levels (8–8.5) well [7,8,12,19] but are sensitive to root asphyxia in heavy soils [22]. Mahaleb seedlings (C 500) proved to be tolerant to calcareous soils and soils with high pH in the NW provinces of China, where during the rainy season in the summer, anaerobic conditions may cause iron chlorosis when using *P. pseudocerasus* as a rootstock [23,24]. Mahaleb seedlings also tolerate *Agrobacterium*-caused crown-gall [6,24].

The effect of *P. mahaleb* rootstocks on tree vigor can range from standard to medium vigor [5], [12–15], [25–30]. Medium vigor for clonal Mahaleb rootstocks is reported for UCMH 59 [15] and IK M9 [13,14]. The only clonal Mahaleb in commercial production is the SL (Saint Lucie) 64 [31], which proved to be vigorous; however, even more vigorous trees are grown from some Mahaleb seedlings [29] or on clonal plums such as Adara, Marianna 2624, and Mayor [19,20,22]. The Mahaleb derivative MaxMa 14 proved to be a moderate-vigorous rootstock in many tests [7,9,19,20]. Rootstocks can also affect the branching angle. In [25,26,30], the authors observed that scions on *P. mahaleb* Magyar showed a large crotch angle, while on MxM 14 and MxM 97, the crotch angle was narrower. Cold hardiness is an important attribute of rootstocks, and rootstocks can affect the response of the scion to cold temperatures. Mahaleb is hardier than Mazzard, and within the Mahaleb species, the broad-leaved subspecies are hardier than the small-leaved subspecies [30].

Researchers agree that rootstocks affect fruit quality; however, their effect on different quality parameters is not consistent. The majority of authors reported larger fruit size and fruit firmness of trees on SL 64 rootstocks or on MaxMa 14, compared to dwarfing rootstocks [8,9,20]; however, others [22,30,32] reported smaller fruit size on MaxMa 14 rootstock. Researchers [33] found higher flower frost tolerance and yield of 'Carmen' on MaxMa 14, while trees on Saint Lucie 64 produced firmer cherries, and on MaxMa 60 rootstock, fruits were darker and sweeter [10]. Interesting correlations were found among quality parameters, such as the positive correlation showed by solid soluble content (SSC) with fruit weight and titratable acid (TA) content [20].

The research on different clonal Mahalebs in Hungary [21,30] resulted in three clonal Mahaleb rootstocks (Bogdány, Egervár, and Magyar) registered and patented by the Hungarian authorities in 2014. Their major advantages are uniformity and a wider vigor range compared to the seedling rootstock varieties. A previous report in 2019 [29] found that the rootstock Bogdány is vigorous (95–110%), Egervár (80–85%), and Magyar (75–80%) showed moderate vigor tendency, while Magyar proved to be most productive in term of yield per tree. Further testing of varieties and rootstocks is important under different site conditions, with the most planted varieties, and under modern spacing and training systems. Based on experiences in earlier test orchards, we now recommend 5 * 1.6 m spacing with taller (4–4.5 m) trees for Hungarian Cherry Spindle [21,34].

In this paper, we present the first 8 years of results of next-stage testing in an orchard planted in optimal site conditions in Ravazd (Hungary), where trees were planted to high-density spacing trained to the modern pruning principles of Hungarian Cherry Spindle [21,34].

## 2. Materials and Methods

Four sweet cherry cultivars were tested in the trial: 'Rita'[Ⓟ], 'Carmen'[Ⓟ], and 'Vera'[Ⓟ], and for pollination, the self-fertile 'Paulus'[Ⓟ]. Trees were budded 10 cm above soil level

on clonal Mahaleb (*Prunus mahaleb* L.) rootstocks selected at the Faculty of Horticulture of MATE Budapest. Clonal Mahalebs are Bogdány, Magyar, and SM 11/4 [12,30]. As a control, rootstock SL 64 (*P. mahaleb*) [31] was used, and MaxMa 14 was used for comparison [35]. Growth, vigor, and productivity of trees were compared to those budded on SL 64 as control. Trees were planted in randomized plots containing three trees each, replicated five times. Scion and rootstock combinations were grouped by varieties, and plots of combinations within rows were randomly planted (random block design). Trees were planted in the spring of 2015, trained to Hungarian Cherry Spindle, and not pruned after planting. From 2016, upright branches were pruned using Brunner's double pruning [21]. The orchard was rain-fed and not irrigated until 2022, and no crop-load measures were applied.

**Tested varieties.** 'Carmen' is characterized as a semi-upright moderate-vigorous tree with medium to good productivity, while 'Vera' is a semi-upright moderate-vigorous tree with good productivity. 'Paulus' is described by [36] as a moderate-vigorous and productive tree recommended for home gardens [37,38]. 'Rita' is moderately vigorous, productive, and ripens early, 7–10 days before 'Burlat' [37,38] (Figure 1).

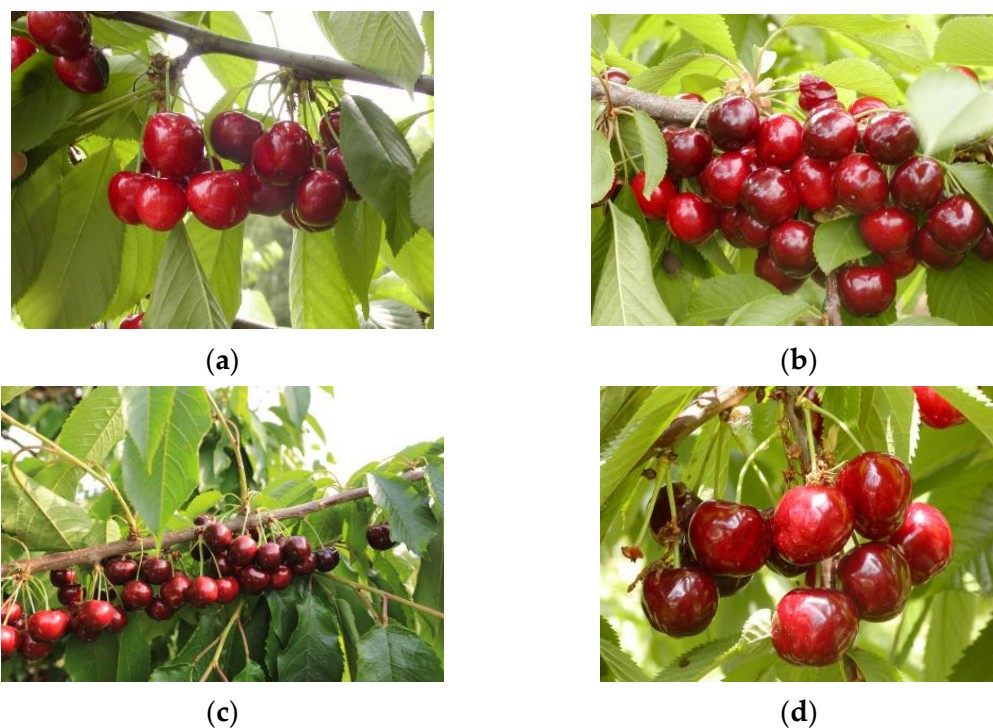

(**a**)   (**b**)

(**c**)   (**d**)

**Figure 1.** The investigated varieties: (**a**) 'Carmen'; (**b**) 'Vera'; (**c**) 'Paulus'; (**d**) 'Rita'.

**Site conditions.** The trial was planted on a private farm in Ravazd, on high-quality soil, on loess developed slightly argillic brown earth, with close neutral pH (KCl) = 7.15, total lime content in the top 60 cm layer 4.3%, and humus content 2.41%. The latitude of the orchard location (47°52′29.55″ N 17°74′39.12″ E) is 225 m on a slight slope in a south–southeast direction. This district of the settlement of Ravazd is classified as vineyard, which indicates the earlier and partly actual land use. The test orchard is part of a private farm maintained by the usual technology in plant protection and fertilizer application. The grass was mown in the alleyways. The spacing was 1.6 m in the row and 5 m between the rows, and tree height was restricted at 4.5 m by heading. From 2016 onward, summer pruning was applied (Figure 2).

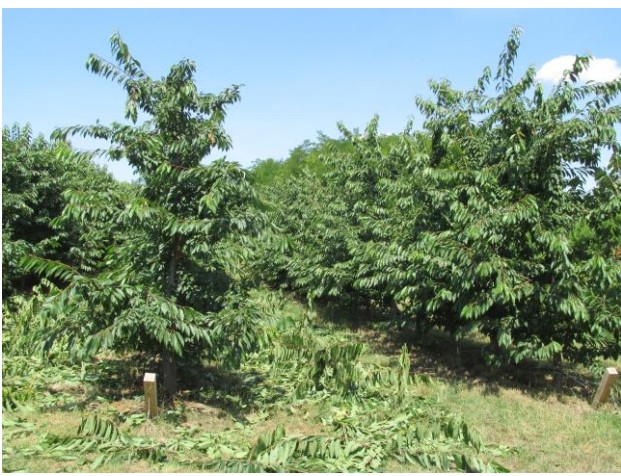

**Figure 2.** The orchard in fifth leaf (2019) after summer pruning.

**Meteorological conditions** of the site, see Table 1 [39]. The average yearly temperature and the number of sunny hours compared to the mean of the last 50 years are increasing, while the average yearly precipitation and the number of rainy days are decreasing. The year 2022 was extremely dry; the precipitation in the first half of the year was 116 mm, 46% of the mean of the last 30 years (253 mm). Additional drip irrigation added 90 mm of water supply.

**Table 1.** Meteorological data during the data collection 2015–2018 [39].

| Parameters | Value |
|---|---|
| Average yearly temperature | 11.8 °C |
| Average yearly temperature during the growing season (between April and September) | 18.4 °C |
| Average yearly precipitation | 595 mm |
| Number of rainy days | 119 |
| Yearly mean number of sunny hours | 2161 |

**Data collection.** The spacing was 1.6 m in the row and 5 m between the rows, and tree height was restricted at 4.5 m by heading. Our database contains data on trunk diameter, canopy size, estimated yield, and fruit weight. Trunk diameter was measured at 50 cm above the graft union on all trees. The canopy width (across the row) and canopy length (parallel to row direction) at the canopy base were also measured in dormant winter conditions.

From the above data, we calculated the following:

- trunk cross-sectional area, TCSA in $cm^2 = \frac{1}{2}$ trunk diameter$^2$ * $\pi$;
- canopy projection area in $m^2$, CA = ($\frac{1}{4}$ (canopy length + canopy width))$^2$ * $\pi$.

Trees began bearing in 2017, the third leaf of the orchard. The yield of three sample trees typical to the variety was measured after harvest. The yield of the remaining 12 trees was estimated and compared to the yield of the measured sample trees. During harvesting, 100 fruits of rootstock–scion combinations were picked randomly for laboratory investigations. In 2020, because of late spring frosts, there were no or very few crops on trees; these years are not included in the cumulated yield calculation. Accordingly, the CY (kg/tree) was evaluated for 5 years and calculated as the sum of the yield of trees from 2017 to 2022, except for 2020. The cumulative yield efficiency index (CYE, kg/cm$^2$, related to TCSA) was calculated as cumulative yield between 2017 and 2022 divided by TCSA (cm$^2$) measured in 2020. For comparison of the precocity of trees, the CY of the first three cropping years (2017–2019) was applied.

**Fruit quality evaluation.** Fruit quality attributes were investigated on three scion varieties, 'Carmen', 'Vera', and 'Paulus' in 2022. The weight of 50 fruits (10 fruits in 5 replicates) per sample of rootstock–variety combinations were examined. The weight of the tested fruits (MFW) and the stone weight (MSW) were measured on a digital scale (KPZ-2-05-4/6000, Klaus-Peter Zander GmbH, Hamburg, Germany). Flesh firmness (FF) of the fruits was determined with a CT3 Brookfield Texture Analyzer (Brookfield Engineering Laboratories, Middleborough, MA, USA) on TA-RT-KIT baseboard using TA 9 pin probe body (test type: TPA, target type: distance, trigger load: 4.0 g, test speed: 1 mm/s, target value: 10.0 mm). Ten fruits per rootstock–variety combination were used for examining fruit flesh firmness. For evaluation of measurement data, TexturePro CT V1.2 Build 9. Software (Brookfield Engineering Lab., USA) was used. The solid soluble content (SSC) of the homogeneous, filtered juice of 15 fruits was determined in $^0$Brix (g 100 g$^{-1}$) with a digital refractometer (ATAGO Palette PR-10, Atago Co., Ltd., Tokyo, Japan) according to Codex Alimentarius 3-1-558/93 [40]. The titratable acid content (TAC) was determined in accordance with the Hungarian standard [41]. TAC content of 3 samples of homogeneous filtered juice, each prepared from 10 fruits of each rootstock–scion combination was determined in 4 replicated measurements. The total acid content (m/m%) was given in malic acid equivalents. The ripening index (RI) was calculated from the ratio of the water-soluble dry matter content to the titratable acid content [20].

**Data analysis.** Data evaluation was carried out with the PASW 18 program (SPSS Inc., Chicago, IL, USA). Analysis of variance (ANOVA) was performed on both factors (varieties and rootstocks) and interactions. When the *F* test was significant, the means of variables (TCSA, CA, CY, YE, MFW, MSW, SSC, TAC, RI, FF) were separated by Tukey's homogeneity test at $p \leq 0.05$ using the IBM SPSS 20 software package.

## 3. Results

### 3.1. Survival of Trees

In the eighth leaf of the orchard, a few trees of the planted 15 individuals were lost. One tree of 'Vera' died until the fourth leaf on each rootstock, SL 64, MaxMa 14, and Magyar, while three trees were lost on the Bogdány rootstock. There are dead trees of 'Rita' on Magyar and MaxMa 14, one tree on each, and two trees on the SL64 rootstock. In all the other rootstock–scion combinations, the planted trees were alive and healthy in the eighth leaf.

### 3.2. Growth of Trees

The statistical analysis proved significant differences between rootstocks; however, no significant differences are indicated between varieties (Table 2). The largest TCSA was measured on 'Paulus' trees, followed by 'Rita', while the TCSA was significantly smaller on trees 'Carmen' and 'Vera' without significant differences. Compared to the control trees (on SL 64), the TCSA of 'Carmen' on Bogdány, 'Paulus' on SM 11/4, and 'Rita' on Bogdány and SM 11/4 was significantly larger. Significant smaller TCSA compared to control SL 64 was measured on MaxMa 14 rootstock with each variety.

**Table 2.** Growth parameters of trees TCSA (cm$^2$) in 2020 and canopy area CA (m$^2$) in 2018.

| Rootstock | Carmen TCSA | | CA | | Vera TCSA | | CA | | Paulus TCSA | | CA | | Rita TCSA | | CA | |
|---|---|---|---|---|---|---|---|---|---|---|---|---|---|---|---|---|
| Bogdány | 119.8 | c | 2.8 | a | 114.0 | b | 3.3 | a | 105.7 | b | 3.2 | a | 123.9 | c | 3.5 | a |
| Magyar | 98.8 | b | 2.5 | a | 92.4 | a | 3.0 | a | 113.8 | b | 3.3 | a | 107.2 | b | 3.5 | a |
| MaxMa 14 | 80.1 | a | 2.5 | a | 93.4 | a | 3.0 | a | 93.4 | a | 2.9 | a | 86.2 | a | 3.3 | a |
| SL 64 | 102.7 | b | 2.2 | a | 105.3 | ab | 1.9 | a | 116.2 | b | 2.9 | a | 109.3 | b | 3.1 | a |
| SM 11/4 | 107.3 | bc | 2.4 | a | 106.3 | ab | 2.1 | a | 131.6 | c | 2.9 | a | 124.1 | c | 3.0 | a |
| *Mean* | *101.7* | *A* | *2.5* | *A* | *102.3* | *A* | *2.7* | *AB* | *112.1* | *A* | *3.4* | *BC* | *110.2* | *A* | *3.3* | *C* |

Note: Means are separated by Tukey's test, values with different letters indicate significant differences at $p \leq 0.05$. Upper case letters in the mean row stand for differences between varieties.

'Carmen' and 'Vera' trees on Bogdány rootstock achieved the largest TCSA, while the TCSA of 'Paulus' and 'Rita' was largest on SM 11/4 rootstock (Table 2). Compared to the largest TCSA-producing rootstocks, trees on Magyar produced significantly smaller TCSA with each variety; however, the difference to MaxMa 14 was significant in the case of 'Vera' only.

Varieties 'Carmen' and 'Vera' showed smaller CA (Table 2) of trees (2.5 and 2.7 m$^2$) compared to 'Paulus' (3.4 m$^2$) and 'Rita' (3.3 m$^2$). The CA of trees on different rootstocks in 2018 did not show significant differences. The CA of 'Carmen' trees was 2.2–2.8 m$^2$, the CA of 'Vera' trees was between 1.9 and 3.3 m$^2$, while trees of 'Paulus' and 'Rita' showed a slightly larger canopy area (2.9–3.3 m$^2$ and 3.0–3.5 m$^2$, respectively).

### 3.3. Cumulative Yield (CY) of Trees between 2017 and 2022

The statistical analysis proved significant differences between varieties and rootstocks within varieties. The CY of trees (kg/tree) by varieties on the investigated rootstocks are displayed in Table 3. 'Carmen' and 'Vera' produced significantly larger CY (38.52 and 36.64 kg/tree) compared to 'Paulus' (19.84 kg/tree) and 'Rita' (22.42 kg/tree).

**Table 3.** Cumulative yield of trees 2017–2022 (kg/tree).

| Rootstock | Carmen | | Vera | | Paulus | | Rita | |
|---|---|---|---|---|---|---|---|---|
| Bogdány | 38.63 | bc | 30.42 | a | 35.68 | c | 21.37 | a |
| Magyar | 42.68 | c | 39.50 | bc | 20.38 | b | 25.17 | b |
| MaxMa 14 | 43.30 | c | 42.04 | c | 16.66 | a | 26.84 | b |
| SL 64 | 33.36 | a | 34.72 | ab | 13.52 | a | 18.78 | a |
| SM 11/4 | 34.63 | ab | 36.55 | b | 12.94 | a | 19.87 | a |
| *Mean* | *38.52* | *B* | *36.64* | *B* | *19.84* | *A* | *22.42* | *A* |

Note: Means are separated by Tukey's test, values with different letters indicate significant differences at $p \leq 0.05$. Upper case letters in the mean row stand for differences between varieties.

The largest cumulative yield (38.63–43.3 kg/tree) was produced by 'Carmen' trees on MaxMa 14, Magyar, and Bogdány, significantly larger than on control SL 64 and vigorous SM 11/4 rootstocks. 'Vera' produced the largest yield on Magyar (39.5 kg/tree) and MaxMa 14 (42.04 kg/tree) rootstocks; in this last one, CY was larger than that of control trees on SL 64. The largest CY of 'Paulus' trees was produced on Bogdány rootstock (35.68 kg/tree), followed by trees on Magyar rootstock. Trees of both clonal Mahaleb produced significantly higher CY than that of control SL 64 (13.52 kg/tree). The CY of 'Rita' trees was largest on MaxMa 14 and Magyar (26.84 and 25.17 kg/tree) rootstocks without significant differences but larger than control SL 64. The trees on other rootstocks yielded considerably less.

### 3.4. Performance of Cumulative Yield Efficiency Index of Trees

In the cumulative yield efficiency index (CYE) calculated to the TCSA, both varieties and rootstocks showed significant differences (Table 4). The highest CYE was recorded on 'Carmen' trees (0.40 kg/cm$^2$), followed by 'Vera' (0.37 kg/cm$^2$) without any significant differences, while the CYE was lower on trees of 'Paulus' (0.18 kg/cm$^2$) and 'Rita' (0.21 kg/cm$^2$).

**Table 4.** The calculated CYE of trees related to TCSA (kg/cm$^2$).

| Rootstock | Carmen | | Vera | | Paulus | | Rita | |
|---|---|---|---|---|---|---|---|---|
| Bogdány | 0.32 | a | 0.27 | a | 0.34 | c | 0.17 | a |
| Magyar | 0.54 | c | 0.43 | c | 0.18 | b | 0.24 | b |
| MaxMa 14 | 0.43 | b | 0.46 | c | 0.18 | b | 0.31 | c |
| SL 64 | 0.32 | a | 0.33 | ab | 0.12 | a | 0.17 | a |
| SM 11/4 | 0.39 | ab | 0.35 | b | 0.10 | a | 0.16 | a |
| *Mean* | *0.40* | *B* | *0.37* | *B* | *0.18* | *A* | *0.21* | *A* |

Note: Means are separated by Tukey's test, values with different letters indicate significant differences at $p \leq 0.05$. Upper case letters in the mean row stand for differences between varieties.

'Carmen' showed the highest CYE on Magyar rootstock, followed by the significantly lower CYE of trees on MaxMa 14 and SM 11/4. Significantly lower CYE was produced from trees on SL 64 and Bogdány. The highest CYE was measured in 'Vera' trees on MaxMa 14 and Magyar rootstock without any significant differences. These rootstocks followed the SM 11/4 and the control SL 64, while the lowest CYE produced trees on Bogdány rootstock. In contrast, 'Paulus' trees produced the highest CYE on Bogdány followed by Magyar and Maxma 14, without any significant differences between the last two rootstocks, while the least productive trees were on control SL 64 and SM 11/4 rootstocks. 'Rita' trees were most productive on MaxMa 14 rootstock, followed by trees on Magyar, while the least CYE was calculated on Bogdány, SL 64, and SM 11/4 rootstocks.

*3.5. Performance of Yielding of Trees 2017 to 2022 and the Precocity*

The first considerable crop on trees occurred in the third leaf (2017) of trees with significant differences between the varieties and rootstocks (Figure 3).

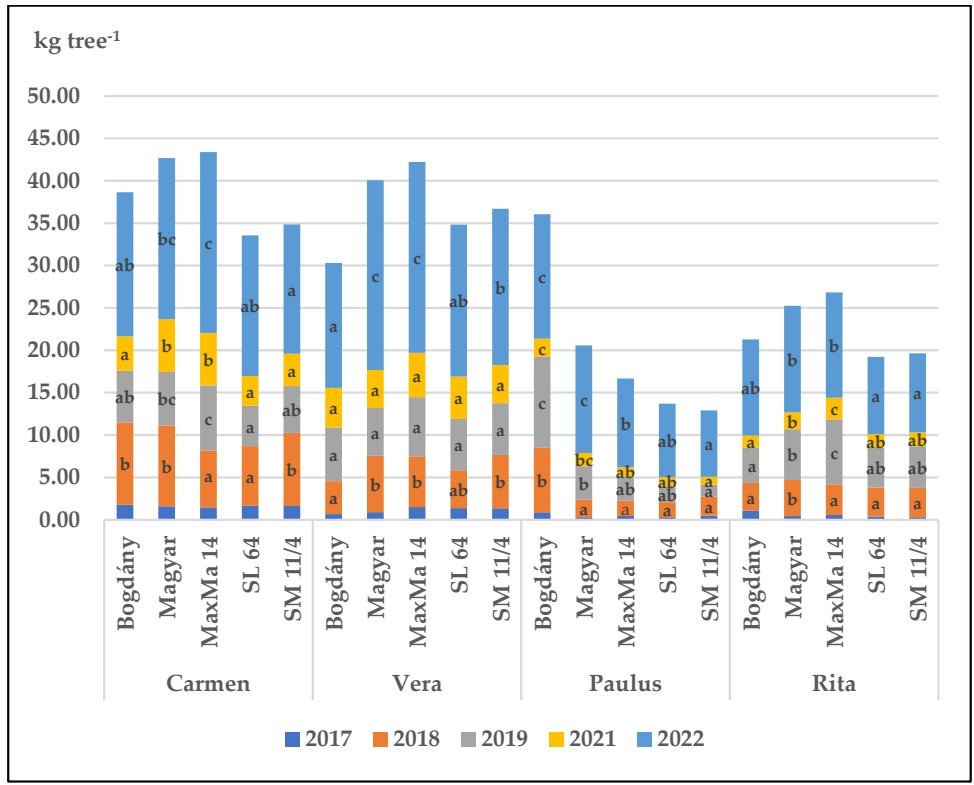

**Figure 3.** Cropping of trees from the third to the sixth leaf of trees (2017 to 2022). Means are separated by Tukey's test; values with different letters within variety indicate significant differences at $p \leq 0.05$.

The differences and cropping tendencies of trees on different rootstocks are the same or very similar to the results of the CY of 2022 (Table 3). In 2020 there was a strong flower frost injury, which resulted in few flowers and fruits; thus, the data from this year are not presented. In the year 2021, the orchard was injured by flower frost damage again, which resulted in a moderate crop (Figure 3). Data in Figure 3 suggest a separate evaluation of the CY of trees from 2017 to 2019 to characterize the precocity.

### 3.6. Precocity of Trees on Different Rootstocks

To characterize the precocity of varieties and rootstocks, the cumulated yield of the first three cropping years from 2017 to 2019 is presented in Table 5. The statistical analysis proved significant differences between varieties and rootstocks. The cumulative yield in the first three cropping years was the highest in 'Carmen' (20.02 kg/tree), 'Vera' was significantly smaller (13.07 kg/tree), while 'Paulus' and 'Rita' produced the least crops (7.82 and 6.46 kg/tree, respectively). 'Carmen' trees on Bogdány rootstock produced a significantly higher yield than that on control SL 64, although the yield of trees on Magyar, SM 11/4, and MaxMa 14 rootstocks was between them. The yield of 'Vera' trees in the first three cropping years was highest on Magyar rootstocks, significantly higher than on SL 64, while on MaxMa 14, SM 11/4, and Bogdány, the yield was in between them. 'Paulus' trees produced outstanding yield in the years 2017–2019 on Bogdány rootstocks, significantly higher than that of trees on other rootstocks. 'Rita' trees produced the highest yield in the first three cropping years on MaxMa 14 rootstocks without any significant differences compared to Magyar and Bogdány rootstocks, while the smallest crop was produced on rootstocks SM 11/4 and SL 64, significantly different from the first group.

**Table 5.** Cumulative yield of trees (kg/tree) in the first three cropping years (2017–2019).

| Rootstock | Carmen | | Vera | | Paulus | | Rita | |
|---|---|---|---|---|---|---|---|---|
| Bogdány | 22.79 | b | 12.31 | ab | 19.17 | b | 6.31 | ab |
| Magyar | 21.17 | ab | 15.00 | b | 6.46 | a | 6.73 | ab |
| MaxMa 14 | 18.24 | ab | 14.50 | b | 5.16 | a | 8.49 | b |
| SL 64 | 17.88 | a | 10.68 | a | 4.10 | a | 5.38 | a |
| SM 11/4 | 20.01 | ab | 12.85 | ab | 4.20 | a | 5.41 | a |
| *Mean* | *20.02* | *C* | *13.07* | *B* | *7.82* | *A* | *6.46* | *A* |

Note: Means are separated by Tukey's test, values with different letters indicate significant differences at $p \leq 0.05$. Upper case letters in the mean row stand for differences between varieties.

### 3.7. Cropping Potential in the Full Crop Year 2022

In the year 2022, we had a full crop without any serious frost injury, and the implemented irrigation helped to overcome the negative effect of the extremely dry season. Table 6 presents the yield of trees in 2022 and the calculated yield for 1 ha (1250 trees/ha). Both 'Carmen' and 'Vera' produced significantly higher yields (17.8 and 19.2 kg tree$^{-1}$) compared to 'Paulus' and 'Rita' (10.9 and 11.1 kg tree$^{-1}$) (Table 6).

**Table 6.** Calculated cropping capacity of trees in 2022 (full crop year).

| | Carmen | | | Vera | | | Paulus | | | Rita | | |
|---|---|---|---|---|---|---|---|---|---|---|---|---|
| Rootstock | kg/tree | | t/ha | kg/tree | | t/ha | kg/tree | | t/ha | kg/tree | | t/ha |
| Bogdány | 17.0 | ab | 21.2 | 14.7 | a | 18.4 | 14.7 | c | 18.4 | 11.3 | ab | 14.2 |
| Magyar | 19.0 | bc | 23.8 | 22.4 | c | 28.0 | 12.7 | c | 15.9 | 12.6 | b | 15.7 |
| MaxMa 14 | 21.3 | c | 26.7 | 22.5 | c | 28.2 | 10.5 | b | 13.1 | 12.4 | b | 15.5 |
| SL 64 | 16.6 | ab | 20.8 | 17.9 | ab | 22.4 | 8.7 | ab | 10.8 | 9.1 | a | 11.4 |
| SM 11/4 | 15.3 | a | 19.1 | 18.4 | b | 23.0 | 7.8 | a | 9.8 | 9.3 | a | 11.7 |
| *Mean* | *17.8* | *B* | *22.3* | *19.2* | *B* | *24.0* | *10.9* | *A* | *13.6* | *11.0* | *A* | *13.7* |

Note: Means are separated by Tukey's test, values with different letters indicate significant differences at $p \leq 0.05$. Upper case letters in the mean row stand for differences between varieties.

'Carmen' trees produced a significantly higher yield on MaxMa 14 compared to control SL 64, followed by Magyar rootstock without significant differences between them, while the yield on SM 11/4 was significantly lower. The highest crop of 'Vera' produced trees on MaxMa 14 and Magyar rootstocks without significant differences between them, while the yield on the other rootstocks was significantly lower. The highest crop of 'Paulus' was produced on Bogdány and Magyar rootstocks, considerably higher than the crop on other rootstocks. The highest yield of 'Rita' produced trees on Magyar rootstocks followed by MaxMa 14 and Bogdány, without any significant differences, while the crop of trees on the other two rootstocks was significantly smaller.

### 3.8. Fruit Characteristics of Varieties on Different Rootstocks

The mean fruit weight (MFW) was the largest in 'Carmen' (10.74 g), the other two investigated varieties produced smaller fruits (Table 7). Within the varieties of 'Carmen' and 'Vera', there were no significant differences between the trees on different rootstocks, while the MFW of 'Paulus' was significantly larger on control rootstock SL64 compared to Bogdány and Magyar.

**Table 7.** Fruit characteristics of varieties on different rootstocks in 2022.

| Rootstocks | MFW (g) | | MSW (g) | | SSC Brix° | | TAC | | RI | | FF (g) | |
|---|---|---|---|---|---|---|---|---|---|---|---|---|
| | | | | | Carmen | | | | | | | |
| Bogdány | 11.00 | a | 0.71 | a | 13.81 | ab | 0.66 | a | 21.0 | d | 27.7 | a |
| Magyar | 11.45 | a | 0.68 | a | 13.57 | a | 0.84 | c | 18.5 | c | 29.9 | a |
| MaxMa 14 | 10.73 | a | 0.69 | a | 14.09 | b | 0.76 | b | 16.1 | a | 30.3 | a |
| SL64 | 10.43 | a | 0.70 | a | 13.56 | a | 0.77 | b | 17.6 | b | 29.8 | a |
| SM 11/4 | 10.11 | a | 0.72 | a | 12.33 | a | 0.65 | a | 19.0 | cd | 32.2 | a |
| *Mean* | *10.74* | *B* | *0.70* | *B* | *13.47* | *B* | *0.74* | *AB* | *18.4* | *B* | *30.2* | *A* |
| | | | | | Vera | | | | | | | |
| Bogdány | 9.58 | a | 0.58 | b | 12.91 | b | 1.19 | c | 10.8 | a | 37.9 | a |
| Magyar | 8.03 | a | 0.52 | a | 11.03 | ab | 0.87 | ab | 15.4 | c | 26.4 | a |
| MaxMa 14 | 8.13 | a | 0.57 | ab | 11.20 | a | 0.73 | a | 12.7 | b | 34.5 | a |
| SL64 | 8.80 | a | 0.53 | ab | 12.54 | b | 0.93 | b | 13.5 | bc | 25.0 | a |
| SM 11/4 | 9.61 | a | 0.57 | ab | 12.14 | ab | 0.75 | abc | 16.2 | d | 30.1 | a |
| *Mean* | *8.83* | *A* | *0.55* | *A* | *11.96* | *A* | *0.89* | *b* | *13.7* | *A* | *30.8* | *A* |
| | | | | | Paulus | | | | | | | |
| Bogdány | 8.07 | a | 0.52 | a | 13.91 | c | 0.53 | a | 23.7 | c | 31.5 | b |
| Magyar | 7.87 | a | 0.56 | a | 12.72 | ab | 0.63 | abc | 18.6 | a | 25.4 | a |
| MaxMa 14 | 8.40 | ab | 0.71 | a | 13.80 | c | 0.74 | c | 20.1 | ab | 31.2 | b |
| SL64 | 9.47 | b | 0.57 | a | 13.59 | c | 0.73 | bc | 19.5 | ab | 32.5 | b |
| SM 11/4 | 8.73 | ab | 0.83 | a | 11.32 | a | 0.54 | a | 20.9 | b | 25.3 | a |
| *Mean* | *8.51* | *A* | *0.64* | *AB* | *13.07* | *AB* | *0.63* | *A* | *20.5* | *B* | *29.2* | *A* |

Note: Means are separated by Tukey's test, values with different letters indicate significant differences at $p \leq 0.05$ within varieties. Upper case letters in the mean rows stand for differences between varieties.

The mean stone weight (MSW) showed significant differences between the varieties 'Carmen' (0.7 g) and 'Vera' (0.55 g). In the stone weight, significant differences were within 'Vera' on Magyar (0.52 g) and Bogdány (0.58 g) rootstocks. There were no significant differences found between rootstocks in MSW of fruits of 'Carmen' and 'Paulus' (Table 7).

Significant higher SSC (Brix°) was found in fruits of 'Carmen' and 'Paulus' compared to 'Vera' (Table 7). 'Carmen' fruits showed higher SSC on MaxMa 14 rootstock compared to control SL 64, Magyar, and SL 11/4. Fruits of 'Vera' showed lower SSC when harvested from trees on MaxMa 14, compared to control SL 64 and Bogdány. The high SSC of fruits of 'Paulus' of trees on Bogdány, MaxMa 14, and SL64 differed significantly from fruits harvested from trees on SM 11/4 rootstock.

The titratable acid content (TAC) of fruits was highest for 'Vera' and lowest for 'Paulus', while the TAC of fruits of 'Carmen' was between them (Table 7). The highest acid content in 'Carmen' fruits produced trees on Magyar rootstock, followed by MaxMa 14 and SL 64, while significantly lower acidity showed fruits harvested from trees on Bogdány and SM 11/4 rootstocks. The highest acid content was found in fruits of 'Vera' from trees on Bogdány, compared to control SL 64, while the least acidity was measured in fruits from trees on MaxMa 14. Significant lower TAC was found in fruits harvested from trees on Bogdány and SM 11/4 rootstock.

The ripening index (RI) was lower in fruits of 'Vera' compared to 'Carmen' and 'Paulus' (Table 7). The RI value of 'Carmen' was significantly higher in fruits on rootstocks Bogdány and Magyar, compared to controls SL 64 and MaxMa 14. The RI of 'Vera' fruits on Magyar rootstock did not differ from the control SL 64, while the fruit on Bogdány was lower, and SM 11/4 had a higher RI. The fruits of 'Paulus' showed a higher RI on Bogdány rootstock compared to control SL 64 and the other tested rootstocks.

No significant differences were found in fruit firmness (FF) of the three varieties, except for 'Paulus', where the fruits from trees on Magyar and SM 11/4 rootstock showed significantly lower firmness values compared to control SL 64, Bogdány, and MaxMa 14.

## 4. Discussion

The trees developed well, according to the Hungarian Cherry Spindle principles. The few tree losses (1 to 3 trees from the planted 15) of 'Vera' and 'Rita' and no lost trees of 'Carmen' and 'Paulus' indicated that the tested varieties do not show early incompatibility with the tested Mahaleb rootstocks.

Our results on tree growth confirmed the previous results [12,25,26,29]. The tree vigor expressed in TCSA performed similarly; however, there are some differences between scion varieties. 'Carmen' and 'Rita' trees on Bogdány developed larger TCSA than that of control SL 64, which corresponds to its vigor. Considering TCSA development, SM 11/4 can be classified as vigorous also, like SL 64 (Table 2). Compared to the control SL 64, only MaxMa 14 rootstock resulted in a smaller TCSA of trees, corresponding to its moderate (80%) vigor [7,9,19,20,32]. As the trees of 'Carmen' and 'Vera' on Magyar rootstock in TCSA showed 81–82% vigor, and the trees of 'Paulus' and 'Rita' 86% vigor, we can conclude in correspondence to our previous results [29] that this rootstock is moderate-vigorous, like MaxMa 14. This trial was the first to provide a comparison of Magyar and MaxMa 14 rootstocks. Based on the growth results, we can conclude that Bogdány rootstock is vigorous, and in varieties 'Carmen', 'Vera', and 'Rita', its vigor exceeds SL64. Rootstock SM 11/4 proved to be vigorous, also. The vigor of trees on MaxMa 14 is the least, 79–82%, compared to the control SL 64. Similar TCSA produced the less vigorous 'Carmen' and 'Vera' trees on Magyar rootstock; however, the slightly more vigorous varieties 'Paulus' and 'Rita' on Magyar rootstock were stronger.

The trees in the fourth leaf were headed at the height of 4–4.5 m, and the CA also filled the space allotted by spacing 1.6 m (2 m$^2$). Calculating on 25% overlapped CA (1 m canopy radius = 3.1 m$^2$/tree CA), the measured CA (Table 2) of trees (2.5–3.4 m$^2$) filled up the space of fruiting branches. From this year onward, regular summer pruning limited the growth of CA, keeping the fruiting branches within the 3.1 m$^2$/tree. The differences in the fourth leaf correspond to the moderate vigor of the varieties 'Carmen' and 'Vera' [37].

Our results on a CY basis (Table 3) confirmed the higher productivity of 'Carmen' and 'Vera' varieties compared to varieties 'Paulus' and 'Rita', which is in correspondence with previous data [29]. Our results support the positive effect of Magyar clonal Mahaleb rootstocks on the productivity of varieties by CY values 'Carmen', 'Vera', 'Paulus', and 'Rita', while on trees on Bogdány rootstock, a positive effect was detected in varieties 'Carmen' and 'Paulus'. The productivity of these rootstock–scion combinations exceeded the trees on control SL 64. The productivity of Magyar rootstock proved to be similar to MaxMa 14 rootstock. In 2022, the orchard was in its eighth leaf and produced the first full crop without flower frost injury. The yield of 'Vera', 'Paulus', and 'Rita' trees this year

followed the tendency of CY of 2017–2022, which supports our conclusion on the rootstock effect on productivity. We can conclude that trees on Magyar rootstock are more productive than the control SL 64; the increase in productivity may range from 28 to 50%, depending on variety. The rootstock Bogdány increased the productivity (CY 2017–2022) of 'Carmen' by 16% and 'Paulus' by 264%. Data on yield efficiency (Table 4) confirm our statement concerning the Magyar rootstock with all the three investigated varieties, but trees grown on Bogdány were more efficient with 'Paulus' only. The Mahaleb clone SM 11/4 resulted in low productivity of trees considering all tested varieties.

Precocity is an important feature of rootstocks in modern high-density orchards. The cherry rootstock *Prunus mahaleb* is more precocious than Mazzard [5]. Earlier data [29] indicated that even Mahaleb seedlings or clonal Mahalebs resulted in precocity close to dwarfing rootstocks. When comparing the CY of the first three cropping years 'Carmen' and 'Vera' proved more precocious than 'Paulus' and 'Rita'. Based on our results (Table 5), 'Carmen' and 'Paulus' trees on Bogdány rootstock, 'Vera' trees on Magyar and MaxMa 14 rootstock, and 'Rita' trees on MaxMa 14 rootstock proved to be more precocious than trees on control SL 64.

Although we are aware that one year of data (2022) is not enough to draw conclusions on fruit quality attributes, our results indicate interesting tendencies (Table 7). Previous research indicated that the fruit size determines the gross crop value of sweet cherry [29,42,43], and the rootstock vigor and water supply positively contribute to the fruit size [44]. Our results agree with [8,20] that quality attributes depend more on the cultivar than on the rootstock; the scion–rootstock combination may influence the fruit size, acidity, and firmness of sweet cherries. Our data on MFW confirm the known fruit characteristics of varieties and the previous results of [29]. Varieties 'Carmen' and 'Vera' did not show rootstock effect on MFW, while the MFW of 'Paulus' differed by rootstocks. MFW was smaller on productive rootstocks Bogdány and Magyar compared to the control SL 64; this effect may cause a larger crop load [22,43]. The mean stone weight of varieties seemingly followed the fruit size, 'Carmen' showed significantly larger MSW compared to 'Vera'. It is conspicuous that the fruits harvested from trees on Bogdany rootstock had significantly larger MSW than that of trees on Magyar rootstock. The rootstock did not affect the MSW of varieties 'Carmen' and 'Paulus'.

The SSC (Brix°) of fruit flesh performed typically to the varieties, the highest SSC was measured in 'Carmen' fruits, significantly lower SSC was found in 'Vera', while an intermediate level was found in 'Paulus'. Higher SSC was measured in 'Carmen' fruits from trees on MaxMa 14 rootstock compared to control SL 64 and other rootstocks, while 'Vera' fruits had lower SSC harvested from trees on MaxMa 14 rootstock. Fruits of 'Paulus' had lower SSC in fruits harvested from trees on rootstocks Magyar and SM 11/4. The TAC performed in a controversial tendency to SSC: the highest was in 'Vera' fruits, contrary to 'Carmen' and 'Paulus'. Rootstocks significantly affected the acid content of 'Carmen' fruits, Magyar rootstock was the highest compared to other rootstocks. Fruits of 'Vera' showed the highest acidity when harvested from Bogdány rootstock, while 'Paulus' fruits showed the highest acidity when harvested from trees on MaxMa rootstock. The ripening index (RI, sugar/acid content) was lower in' Vera' fruits than 'Carmen' and 'Paulus', which is in agreement with the organoleptic experience: there is more acid taste in 'Vera' fruits. The higher RI value of 'Carmen' on rootstocks Bogdány and Magyar and 'Paulus' on Bogdány rootstock can be considered as an additional advantage of these rootstocks.

The fruit firmness did not show any significant differences between varieties and rootstocks, except for the 'Paulus' fruits, which were less firm on Magyar and SM 11/4 rootstocks. These data indicate that rootstocks may affect these important fruit characteristics; however, further, more detailed investigation is needed to explore all possible factors (harvesting time, ripening, etc.) influencing these characteristics.

Certain observations on growth habits and branching angles were made over the years, which are presented in (Figure 4). Varieties on both rootstocks Bogdány and Magyar showed abundant flat branching, a larger crotch angle, and more fine fruiting wood compared to MaxMa 14, SL64, or SM 11/4.

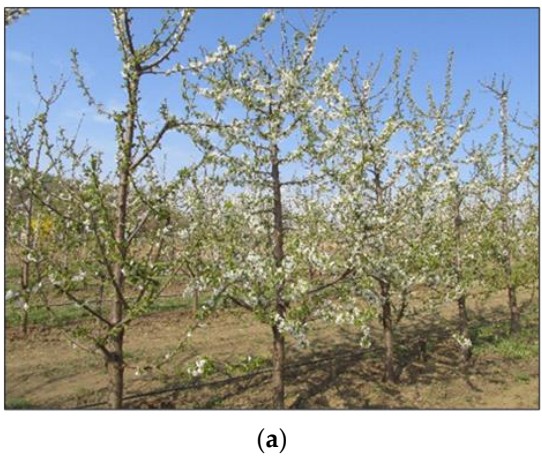
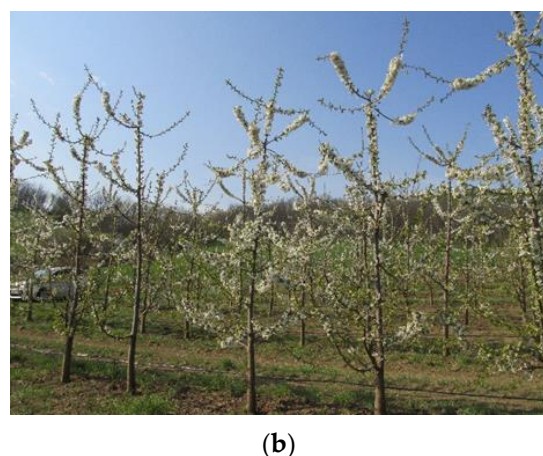

(**a**)                                                              (**b**)

**Figure 4.** Effect of rootstocks on the branching habit of trees: (**a**) 'Paulus': left one tree on SL 64 with upright growth, to the right trees on Bogdány with abundant flat branching; (**b**) two 'Rita' trees left on MaxMa 14, right two trees on Magyar rootstock.

It is worth mentioning that in the year 2022, after harvest, no irrigation was applied, and the trees were kept in rain-fed conditions. In the extreme drought, the trees on MaxMa 14 rootstock showed slight water deficiency, pale green leaves, and light wilting, while trees on other Mahaleb rootstocks did not show these symptoms.

## 5. Conclusions

The rootstock/scion interactions in sweet cherry perform individually to each combination. The Hungarian clonal Mahalebs, Magyar and Bogdány, show advantages in important rootstock features considering vigor, productivity, precocity, and fruit characteristics, which makes them applicable in high-density orchards. Magyar rootstock is moderate-vigorous (about 80%), as is MaxMa 14, but results in abundant and flat branching, precocious trees with high productivity and good fruit size for 'Carmen' and 'Vera'; thus, it is applicable rootstock for high-density orchards. The potential yield of trees on this rootstock in the full crop stage can be expected to be about 12.6–22.4 kg/tree. Bogdány rootstock is vigorous (about 100%) in TCSA, exceeding the SL64, but contrary to SL 64, results in abundant and flat branching, precocious trees with high productivity and good fruit size for 'Carmen' and 'Vera'. The excess crop load may result in smaller fruit size in 'Paulus'. Bogdány is also an applicable rootstock for high-density orchards. The potential yield of trees on this rootstock in the full crop stage can be expected to be about 14.7–17 kg/tree. Our results confirmed that the rootstock features of MaxMa 14 are moderate-vigorous (about 75–80%), with lateral branches of the scion in a narrow crotch angle and upright growing. Trees are precocious, highly productive, and produce good-sized fruit for 'Carmen', 'Vera', and 'Paulus'. In extremely dry summers, they are less tolerant to drought than clonal Mahaleb rootstocks. MaxMa 14 is an applicable rootstock for high-density planting. The potential yield of trees on this rootstock in the full crop stage can be expected to be about 13.1–26.7 kg/tree.

**Author Contributions:** Conceptualization, K.H.; methodology, K.H., L.M. and G.F.; formal analysis L.M. and G.F.; investigation, K.H., L.M. and G.F.; data curation, K.N.-C., G.F. and L.M.; writing—original draft preparation, K.H. and G.F.; visualization, K.H.; supervision, K.H.; All authors have read and agreed to the published version of the manuscript.

**Funding:** This research received no external funding.

**Institutional Review Board Statement:** Not applicable.

**Informed Consent Statement:** Not applicable.

**Data Availability Statement:** Not applicable.

**Acknowledgments:** Authors express special thanks to the farm owner Imre Steczina for keeping the orchard in excellent shape and for technical support of research.

**Conflicts of Interest:** The authors declare no conflict of interest.

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
