# Peer review of "Growth and Productivity of Sweet Cherry Varieties on Hungarian Clonal Prunus mahaleb (L.) Rootstocks"

_horticulturae, doi:10.3390/horticulturae9020198_

Round 1

Reviewer 1 Report

High quality rootstocks are important for cherry production industry. Prunus mahaleb is a major cherry rootstock widely used in the world, but the evaluation on effect of rootstocks of this species and its hybrids on cherry trees’ growth performance and fruit yield is not enough in orchards. Based on four years’ experiments, this study reported the important relationship on cherry’s crop between varieties and rootstocks, which provided a good reference for cherry production in orchards. The experimental is reasonably designed, and the conclusions are supported by the data. However, on MS writing, much revision work should be done to improve language, mainly including

(1)    Low quality writing in many places: For example, Line 29: P. mahaleb is a major cherry rootstock in Central and Southern European countries as.., but in Line 34: Despite of all advantages of Prunus mahaleb still…, Note: when a species name starts a sentence or occurs for the first time, it should be fully spelled.  

Line 63: The effect of P. mahaleb rootstocks on tree vigor…..; Line 64: Medium vigor for clonal mahaleb rootstocks is……; Line 73: Mahaleb is hardier than Mazzard, and within the Mahaleb species the….; Line 84: resulted three clonal Mahaleb rootstocks… Note: these inconsistent spelling forms make the readers very confused.

Line 68: MaxMa 14 but Line 103:MaxMa 14’

Line 41: ……the so called „pedestrian orchards” of… should be “ “.

Line 43: ……density up to 1000 – 5000… but in Line 57: tolerate high lime content and pH level (8-8.5) well. n-dash line not a hyphen should be used for range values.

……..

(2)    Inconsistent verb tenses: Line 286 to 288: ‘Rita’ trees produced the highest yield in the first three cropping year on MaxMa 14 rootstocks without any significant differences compared to Magyar and Bogdány root, while the least crop is produced by rootstocks……

(3)    Inconsistent spelling form/format in many places: For example, Table 2 to 6, for varieties Carmen, Vera, Paulus, and Rita, it is ok when ‘ ‘ is not included, but included in Table 7.

(4)    Some grammar errors, for example: Line 163: The weight of 50 fruits per samples of rootstock;  

Line 188: Line 252: ……leaf on each rootstocks SL 64, MaxMa 14 and Magyar, …while the least YE were calculated on Bogdány, SL 64 and SM 11/4 rootstocks. Line 284: …on MaxMa 14, SM 11/4 and Bogdány the yield was in between them.

(5)    Line 112: Tested varieties. Variety ’Carmen’ is characterized…Note: delete Variety because ‘ ‘ for Carmen already indicates clear on its identity.

(6)    References: Check on if spelling format meet the journal’s standard. And abbreviated journal names should be used for 14, 25, and 33.

(7)    Many other minor revision comments are seen in marked MS, including some unreadable sentences.

(8)    Finally: suggest to change title as “Growth and productivity performance of sweet cherry varieties on Hungarian clonal rootstocks of Prunus mahaleb (L.) and its hybrids”. On experimental design or data collection: more details should be included, such as how to choose three trees for harvesting fruits. On experimental design or data collection: more details should be included, such as how many trees each variety were used for measuring trunk diameter and canopy;  how to choose three trees for harvesting fruits.

Author Response

Reviewer 1

Thank you for your time, reading and evaluating our MS. Thank you for your valuable comments.

on MS writing, much revision work should be done to improve language, mainly including

Line 29: P. mahaleb is a major cherry rootstock in Central and Southern European countries as.., but in Line 34: Despite of all advantages of Prunus mahaleb still…, Note: when a species name starts a sentence or occurs for the first time, it should be fully spelled.

Answer: thank you for your comment, the writing the latin names we corrected in the revised MS.

Line 63: The effect of P. mahaleb rootstocks on tree vigor…..; Line 64: Medium vigor for clonal mahaleb rootstocks is……; Line 73: Mahaleb is hardier than Mazzard, and within the Mahaleb species the….; Line 84: resulted three clonal Mahaleb rootstocks… Note: these inconsistent spelling forms make the readers very confused.

Answer: thank you for your comment, the writing the name “Mahaleb” we corrected in the revised MS.

Line 68: MaxMa 14 but Line 103:MaxMa 14’

Answer: thank you for your comment, the writing the name MaxMa 14 we corrected in the revised MS.

Line 41: ……the so called „pedestrian orchards” of… should be “ “.

Answer: thank you for your comment, the writing “pedestrian orchard” we corrected in the revised MS.

Line 43: ……density up to 1000 – 5000… but in Line 57: tolerate high lime content and pH level (8-8.5) well. n-dash line not a hyphen should be used for range values.

Answer: thank you for your comment, the hyphen is replaced by n-dash line in Line 43.

……..

(2)    Inconsistent verb tenses: Line 286 to 288: ‘Rita’ trees produced the highest yield in the first three cropping year on MaxMa 14 rootstocks without any significant differences compared to Magyar and Bogdány root, while the least crop is produced by rootstocks……

Answer: thank you for your comment, the verb tenses are corrected in revised MS.

(3)    Inconsistent spelling form/format in many places: For example, Table 2 to 6, for varieties Carmen, Vera, Paulus, and Rita, it is ok when ‘ ‘ is not included, but included in Table 7.

Answer: thank you for your comment, the spelling form of variety names in Table 7 is corrected.   

(4)    Some grammar errors, for example:

Line 163: The weight of 50 fruits per samples of rootstock; 

Answer: corrected

Line 188: leaf on each rootstocks SL 64, MaxMa 14 and Magyar,

Answer: corrected

Line 252: ……leaf on each rootstocks SL 64, MaxMa 14 and Magyar, …while the least YE were calculated on Bogdány, SL 64 and SM 11/4 rootstocks.

Answer: corrected

Line 284: …on MaxMa 14, SM 11/4 and Bogdány the yield was in between them.

Answer: corrected

(5)    Line 112: Tested varieties. Variety ’Carmen’ is characterized…Note: delete Variety because ‘ ‘ for Carmen already indicates clear on its identity.

Answer: corrected

(6)    References: Check on if spelling format meet the journal’s standard. And abbreviated journal names should be used for 14, 25, and 33.

Answer: corrected

(7)    Many other minor revision comments are seen in marked MS, including some unreadable sentences.

Answer: Thank you very much for the minor revision comments, majority of them are accepted and corrected in MS.

Answers to not accepted comments: in Line 16 we used the technical term “trained”, in Line 49 “summer pruning” . Both are accepted technical terms in tree training, thus we do not accept your proposal. In Line 56 we use the verb “suite” in term of adaptability.

(8)    Finally: suggest to change title as “Growth and productivity performance of sweet cherry varieties on Hungarian clonal rootstocks of Prunus mahaleb (L.) and its hybrids”. On experimental design or data collection: more details should be included, such as how to choose three trees for harvesting fruits. On experimental design or data collection: more details should be included, such as how many trees each variety were used for measuring trunk diameter and canopy;  how to choose three trees for harvesting fruits.

Answer: Thank you for your suggestion to change the title but we do not accept. Your proposal would make longer the title without any further important information.

You asked more details about the choosing trees for harvesting. It is included, see line 150-152: On each variety the yield of three sample trees typical to the variety was measured after harvest. The yield of the rest 12 trees was estimated, compared them to the yield of the measured sample trees.

You asked, how many trees were used for measuring trunk diameter and canopy: Trunk and canopy was measured on all the trees, 15 trees per rootstock-scion combinations.

In Line 150 you suggest replace “turned to bearing” by “started to bearing. The first technical term “turned to bearing” is widely accepted in pomology, we do not change it.

Reviewer 2 Report

The manuscript reports that due to climate changes, the drought and lime tolerant Prunus mahaleb rootstock may be important. In this sense, the testing of mahaleb rootstocks and hybrids in intensive orchards is still missing. 

The main strength of the work is the results of six seasons, which gives great power to the conclusions, which can be very valuable for cherry growers in Hungary, and regions with similar climatic and soil characteristics. However, the work lacks a scientifically relevant novelty that justifies its publication in Horticulturae. 

Author Response

Reviewer 2

Thank you for your comments that contribute to improvement of our MS.

We are glad to see that you agree that the testing of clonal Prunus mahaleb rootstocks may have attributes, which makes possible their application in future intensive orchards. Due to the climate change and extremely dry summer weather the growers in Central and S – SE-Europe are looking for rootstocks, more vigorous than the widespread planted dwarfing rootstocks. Further on in replant situations also moderate vigorous rootstocks are recommended. We agree with your remarks that our results of six seasons support well the conclusions, thus some of the tested rootstocks could be valuable choice for growers in Hungary and regions with similar climatic and soil characteristics.

However, we do not agree, that the scientific novelty of the paper is missing. Probably we did not emphasize clearly that these rootstocks are first tested in West-Hungary, optimal soil climatical conditions and in comparison with the control SL 64 and the mahaleb hybrid MaxMa 14. In the revised MS we improved this insufficiency.

Thank you again taking time to read and comment of our MS. 

Reviewer 3 Report

The manuscript is very difficult to read and understand. The methodology is not clearly introduced. The results is not stated well. The discussion and the conclusion dont focus on the findings and the significance of the study. It need thorough improvement before further submission.

Author Response

Reviewer 3

Thank you much for your comments. In the revised MS we intended to improve to read and understand the text easier. We described the methodology and results more precise and understandable. In the conclusion we emphasized more the new findings and focus to the significance of the study.

Reviewer 4 Report

This manuscript describes the influence of the rootstock on several agronomic and qualitative parameters. However, qualitative analyses were carried out only in one year. The study design is appropriate, but the presentation of the results needs to be improved, and the format of the introduction, results, discussion, and citations needs to be revised.

The comments of this manuscript are as following:

Line 5: remove the comma

Line 112: were there four varieties?

Line 151: Was the yield estimated on only three trees per variety?

Line 162: were there four varieties? was there also the Rita variety?

Line 162:  Were the qualitative analyzes carried out only in one year (2022)?

Line 163: on line 153 you say that the average weight was calculated on 100 fruits (MFW). How many fruits did you use for the qualitative analyses? And did not use replications?

Table 2: I recommend using the letter “a” to indicate the highest value (You should standardize in the text). Add the standard deviation in table

Line 205: write Bogdány rootstocks and not root. You should standardize the word in the text

Figure 2: is the Cumulative yield described? make the graph clearer. The statistics for the year 2017 are not indicated.

Table 3: Add the standard deviation in table

Line 316: The author could add photos of the four varieties in the qualitative paragraph.

Line 322: The author writes “average weight of stones (MSW)”, but in materials and methods he writes “average seed fruit weight”.

Line 346: Remove the word "sugar/acid content" which is described in materials and methods

Table 4: Is YE the Cumulative Performance Efficiency Index (CYE)? You should standardize the word in the text. Add the standard deviation in table

Table 5: Add the standard deviation in table

Table 6: Add the standard deviation in table

Table 7: expand the caption in the table by adding more information. Add the standard deviation in table.

Author Response

Reviewer 4

Thank you for your comments that considerable contribute to improvement of our MS.

Thank you for your remark that the study design is appropriate but the presentation but the presentation of the results should be improved. In the revised MS we improved the presentation of results, and revised the introduction, results,  and discussion.

Here are the responses to your detailed comments:

Line 5: remove the comma:
Answer: accept, thank you, the comma is removed

Line 112: were there four varieties?:
Answer: The missing description of ‘Rita’ is included: . ‘Rita’ is moderate vigorous, productive, ripen early, 7-10 days before ‘Burlat’ [37,38].

Line 151: Was the yield estimated on only three trees per variety?
Answer: No, we measured the yield on three sample trees, than the yield on the rest of trees we estimated. The corrected sentence: “On each variety the yield of three sample trees was measured after harvest. The yield of the rest 12 trees was estimated, compared them to the yield of the measured sample trees.”

Line 162: were there four varieties? was there also the Rita variety?
Answer: No, the fruit quality attributes are investigated on three varieties. Unfortunately, the ‘Rita’ fruits we could not harvest on time due to technical circumstances.

Line 162:  Were the qualitative analyzes carried out only in one year (2022)? 
Answer: Yes, the qualitative analyzes were carried out in one year only. We planned the qualitative analyzes in full crop years, from 2020 onward. Unfortunately in 2020 the severe flower frost damage (few fruits/tree) and the moderate flower frost damage (1/3rd crop) in 2021 resulted in low crop load, which distorts the quality attributes. That is why the fruit qualitative analyses were carried out in 2022.

Line 163: on line 153 you say that the average weight was calculated on 100 fruits (MFW). How many fruits did you use for the qualitative analyses? And did not use replications?
Answer: Thank you for the comment, we corrected the methodology description of qualitative analyses.  In line 153 the sentence corrected: “During harvesting 100 fruits of rootstock-scion combinations were picked randomly for laboratory investigations.”
The corrected line 164: The weight of 50 fruits (10 fruits in five replicates) per samples of rootstock-variety combinations were examined.
Corrected line 170-171: Ten fruits per rootstock-variety combinations were used for examining fruit flesh firmness.
Corrected line 177-179: added „TAC content of three samples of homogeneous filtered juice, each prepared of 10 fruits of each rootstock-scion combination was determined in four replicated measurements.”

Table 2: You recommend using the letter “a” to indicate the highest value (You should standardize in the text). Add the standard deviation in table.
Answer: Thank you for your recommendation but we do not accept. Concerning the sign (a,b,c etc.) of homogeneous groups, I do not see importance, which letter indicate the highest value. On the other hand, the adding the standard deviation to the tables (from 2 to 7) would make impenetrable and crowded the tables. Our data collected in a field trial (with three trees per plot and five times replicated, which means 15 trees in each combination) supported by ANOVA are reliable. Besides the signing of homogeneous groups (separated by Tukey’s test) I do not see the necessity of adding the standard deviation.

Line 205: write Bogdány rootstocks and not root. You should standardize the word in the text
Answer: Thank you, we replaced the word “root” by “rootstock”

Figure 2: is the Cumulative yield described? make the graph clearer. The statistics for the year 2017 are not indicated.
Yes, we presented the cumulative yield in Table 3. The graph in Figure 2 presents the yearly crop of trees. The yield of trees in the first year (2017) was low, signing the homogeneous groups in the graph would make the Fig 2 crowded, that is why statistics (sign of homogeneous groups) is not indicated in 2017.  

Table 3: Add the standard deviation in table
Answer: see answer Table 2

Line 316: The author could add photos of the four varieties in the qualitative paragraph.
Answer: photos of the investigated four varieties are added in Figure 1, Section Materials and methods, Line 112-113.

Line 322: The author writes “average weight of stones (MSW)”, but in materials and methods he writes “average seed fruit weight”.
Answer: We applied the “mean stone weight (MSW)” in the text of materials and methods and in results section.

Line 346: Remove the word "sugar/acid content" which is described in materials and methods.
Answer: thank you, the word "sugar/acid content" is removed in revised MS

Table 4: Is YE the Cumulative Performance Efficiency Index (CYE)? You should standardize the word in the text. Add the standard deviation in table
Answer: thank you for the comment, the word is standardized in the text. Concerning the standard deviation see the answer at Table 2

Table 5: Add the standard deviation in table. Concerning the standard deviation see the answer at Table 2

Table 6: Add the standard deviation in table. Concerning the standard deviation see the answer at Table 2

Table 7: expand the caption in the table by adding more information. Add the standard deviation in table. Concerning the standard deviation see the answer at Table 2

Round 2

Reviewer 2 Report

Dear Károly Hrotkó, 

I think that you have done a great job improving the first version of MS. 

Author Response

Thank you for your comment "I think that you have done a great job improving the first version of MS." Well, we are grateful to our reviewers, their comments contributed considerably to the improvement.

Well, we agree that the research design could be improved. Due to technical circumstances the fruit quality investigations confined to three varieties, unfortunately this is now impossible to replace.

We checked all the references, by our assesment all of them are relevant to the research topic.

Reviewer 3 Report

The manuscript got a big improvement after revision. There are several minor modifications needed.

Line 80: while [10] found that trees on...were darker and sweeter. can be modified as while trees on...were darker and sweeter[10]. 

Line 87: Bujdosó et al. (2019) [29] found ..., (2019) can be deleted or revised to in 2019

Line 386: ...stocks onto productivity... onto should be on

Author Response

Thank you for your comment: "The manuscript got a big improvement after revision." We are grateful to our reviewers for taking time to find our mistakes in MS. The reviewers considerably contributed to the improvement of th MS.

We accept all of your comments. Your suggestion for improvement in English style in following lines are corrected in the final revised version of our MS.

Line 80: “while [10] found that trees on...were darker and sweeter.” can be modified as “while trees on...were darker and sweeter[10]. ”

Line 87: “Bujdosó et al. (2019) [29] found ...”, (2019) can be deleted or revised to “in 2019”

Line 386: ...stocks onto productivity... onto should be on
